# Beyond Cosine Similarity: Introducing the Unified semantic Similarity Metric Benchmark (USMB) for Text Similarity Measurements

## Abstract

Text embedding models are increasingly utilized in production across various applications, from Information Retrieval (IR) to document parsing, but relatively little research has been focused on how to best utilize these embeddings for downstream tasks. While cosine similarity, a popular measure of embedding and text similarity, is widely used, it may not be the strongest metric choice for all tasks. In this work, we introduce the Unified semantic Similarity Metric Benchmark (USMB), a novel leaderboard for text similarity metrics composed of 5 unique tasks and 30+ datasets with the goal of providing a standardized means of measuring the effectiveness of a text similarity metric on a suite of challenging tasks encompassing the nuances of semantic understanding. Additionally, we demonstrate that while cosine similarity achieves the highest score on our benchmark of any pre-existing metric, developing a task-specific ensembled model using our metrics leads to a 40.3% increase in benchmark performance relative to cosine similarity. We hope that through this work, greater attention can be given to potential performance gains through metric selection and that the field's ability to measure semantic similarity advances as a result.

## 1 Introduction

Text similarity measurements, especially within the context of LLM applications, are a critical component of production-level machine learning systems. The use cases hinging on a proper understanding of text similarity range from consumer considerations such as text summarization and content creation to industries spanning legal documentation (Bhattacharya et al., 2022), patent similarity (Yoo et al., 2023), and scientific writing (Yu et al., 2022). While modern research has typically focused on developing domain-specific models for these tasks (Xu et al., 2024), the text similarity metric used in these tasks is often overlooked and untested, with cosine similarity being the uncontested default (Zhou et al., 2022) despite not necessarily being the best choice for all tasks.

With all these use cases where measuring semantic similarity is paramount, there is a sore lack of exploration into methods that effectively and robustly encompass this broad human concept, and a lack of data to train models themselves towards this objective. To tackle the difficult problem of task-based metric selection, we develop and introduce the Unified semantic Similarity Metric Benchmark (USMB), a systematic evaluation of text similarity metrics on 5 unique tasks and 30+ datasets spanning human preference alignment, transformation robustness, information sensitivity, clustering performance, and retrieval robustness. We measure the performance of popular metrics like cosine similarity, Levenshtein (edit) distance, Rouge score, BM25, and Jaccard similarity over each task and demonstrate that while cosine similarity achieves the highest overall score of these metric by 6.1%, no metric clearly dominates in all tasks considered.

After evaluating pre-existing metrics on each task, we focus on developing a naive ensembled model for each task with the goal of improving performance by taking advantage of each metric's strengths and weaknesses. By simply using an out-of-the-box linear regression or random forest classifier depending on the task and training it on the 5 metrics we measured, we're able to achieve top scores on all categories of the USMB, beating cosine similarities overall score by 40.3%.

We hope that the USMB serves as a helpful tool in metric selection to complement existing benchmarks focused on model selection, and aim to demonstrate the power of ensembling based on the downstream usage of a chosen metric.

## 2 BACKGROUND

### 2.1 ALGORITHMIC TEXT SIMILARITY MEASUREMENTS

For humans, judging the similarity between two pieces of text can be framed as a grade-school-level task. This type of overarching similarity, encompassing tone, writing style, meaning, and factual accuracy, is called *semantic similarity* (Chandrasekaran & Mago, 2021). Human-powered solutions, however, aren't scalable to corpora consisting of tens of thousands of documents, hence the need for machine-powered similarity measurements.

Levenshtein distance is one machine-powered way of measuring similarity that measures the edit (insertion, deletion, substitution) distance between two strings.

Similarly, we can measure similarity through overlap in word frequency with the Jaccard similarity coefficient (Jaccard, 1912), which divides the intersection of the words in the texts by their union.

TF-IDF (Term Frequency-Inverse Document Frequency) is used to evaluate the importance of a word in a document, which itself is part of a corpus. The importance increases proportionally to the number of times a word appears in the document but is offset by the frequency of the word in the corpus (Jones, 1972). TF-IDF is parameterized for a document $d$ with $t$ representing the frequency of a term within the document. $D$ is our corpus of documents.

BM25 (Robertson et al., 1994) is another advanced text similarity metric that builds on top of TF-IDF to rank documents based on a given query. The original BM25, Okapi BM25, modifies TF-IDF by incorporating document length and the frequency of terms in a query (Manning et al., 2009). BM25 can be parametrized for a document $d$ and a query $q$ with $t$ representing a term in the query, $|d|$ as the length of the document $d$, avgdl as the average document length in the corpus, and $b$ and $k_1$ as hyperparameters. We will use BM25+, which adds a small constant $\delta$ as a lower bound.

Finally, we introduce Rouge (Recall-Oriented Understudy for Gisting Evaluation) score (Lin, 2004), particularly useful for evaluating summarization tasks. Rouge measures the overlap of n-grams, word sequences, and word pairs between two documents. We report the average of Rouge-1 and Rouge-2 scores, which measure 1-gram and 2-gram overlap between documents.

There are many similarity metrics we haven't covered for brevity but are still notable, such as BLUE (Papineni et al., 2002), BERTscore (Zhang et al., 2020), and MAUVE score (Pillutla et al., 2021).

All equations and time complexities for the metrics we consider can be found in appendix A and C.

### 2.2 TEXT EMBEDDING MODELS AND COSINE SIMILARITY

To move past algorithmic similarity measures, we need some way to learn the representation of text in a latent space. We do so with text embedding models, which take an arbitrary-length string and transform it into a fixed-length vector. This process is referred to as "embedding" a piece of text.

The output of text embedding models is often interpreted as capturing the semantic meaning of the text passed into them (Senel et al., 2018) in a high-dimensional latent space. This space, geometrically crafted throughout the model training process, has been shown to arrange texts with similar meanings close together, allowing nuanced relationships between words to be discerned based on their proximity and orientation in this space.

Most uses of text embeddings hinge on their ability to represent meaning in a high-level geometric space. Cosine similarity is a popular metric used to measure the similarity between two pieces of text via their embeddings. The equation for cosine similarity can be found in appendix A.

One of the most common use cases of cosine similarity is information retrieval, used for Retrieval-Augmented Generation (RAG) (Lewis et al., 2021) in the context of LLMs. Classification is another application of applying text similarity measures to text embeddings. Some prevalent use cases of clustering are plagiarism detection (Jiffriya et al., 2014) and LLM watermarking (Huo et al., 2024).

## 2.3 USAGES OF TEXT SIMILARITY MEASURES IN BENCHMARKING

The popularity of embedding models has inspired the creation of leaderboards to measure model performance in a standardized manner. A popular leaderboard is the Massive Text Embedding Benchmark (MTEB) (Muennighoff et al., 2023), which utilizes 56 datasets across 8 tasks to assign a final score to an embedding model. This leaderboard builds off of previous, more specialized benchmarks, such as the Benchmark for Zero-shot Evaluation of Information Retrieval Models (BEIR) (Thakur et al., 2021) and Unsupervised Sentence Embedding Benchmark (USEB) (Wang et al., 2021). However, these benchmarks primarily focus on evaluating embedding models rather than the text similarity metrics themselves, and use cosine similarity without any analysis or discussion around this decision. Our work introduces a novel approach to benchmarking by providing an evaluation framework specifically designed to assess and compare text similarity metrics themselves.

## 3 EVALUATING TEXT SIMILARITY MEASUREMENTS ON ROBUSTNESS AND ALIGNMENT

To construct our metric leaderboard, we develop 5 unique tasks meant to assess the effectiveness of a metric on the axes of alignment and robustness. We will start by defining these two concepts.

**Alignment** describes a metric's ability to reflect human judgments and preferences. A similarity metric is considered aligned if it reflects human evaluations regarding text similarity. This is especially important in applications like summarization and content moderation, where the end-users perception and understanding significantly impact the effectiveness of the LLM.

**Robustness**, on the other hand, refers to a metric's resilience against irrelevant or non-semantic modifications in text, such as random capitalization, deletions, or misspellings. A robust similarity metric consistently identifies texts that convey the same meaning, despite superficial changes, and clearly distinguishes texts that differ significantly in content, intent, or meaning.

### 3.1 MODELS

We test various models to demonstrate the consistency of our results across multiple popular embedding architectures. We used six open-source models: BGE-m3 (Chen et al., 2024), E5-Large-V2 (Wang et al., 2022), Nomic-Embed-Text-v1.5 (Nussbaum et al., 2024), E5-RoPE base (Zhu et al., 2024), Jina-Embeddings-v2-Base (Günther et al., 2024), and Mosaic-Bert-Base (sequence length 1024) (Press et al., 2022). We additionally test Cohere's Embed-English-v3.0 (Reimers, 2023) and OpenAI's Text-Embedding-3-Small (OpenAI, 2024) due to their popularity and real-world applicability. Although we selected these models for their diverse architectures and performance, we acknowledge these may not generalize to other models and datasets. Detailed information such as context lengths and parameter counts for these models can be found in Appendix B. For texts exceeding these models' context windows, we truncate from the end of the input.

### 3.2 REPORTING METHODOLOGY

We test 5 pre-existing metrics: cosine similarity, BM25+, Jaccard similarity, Rouge score, and Levenshtein ratio. BM25+ scores are normalized between 0 and 1; other scores are naturally bounded in this range. Cosine similarity scores are calculated for each model and averaged across models. Our primary goal is to introduce the USMB benchmark and demonstrate how cosine similarity as a whole can be combined with more traditional text similarity metrics. We average cosine similarities across models to account for model-based variation in this metric, an attribute which doesn't exist in the other metrics we consider.

When creating ensembled models, we use the following setup. We reserve a random sample of 80% of the data to use as our training data, and report all ensembled model scores on the remaining 20%. We average our results across 1,000 random seeds for each model and task. We use the same conditions to fit and evaluate the best-performing stand-alone metric. We use the default parameters from scikit-learn (Pedregosa et al., 2011). For linear models, we do not fit an intercept term. To reduce the chances of biases and architectural differences between embedding models being a confounding factor in our analysis, we train a separate ensemble model using each embedding model's cosine

similarity output, and run our evaluation on each of these models. We then average these 8 scores (derived from our 8 models) and report this final averaged score as the overall score of the ensemble method. We report the computational complexity of our ensembling methods in appendix C. The contribution of each metric to the final ensemble model for each task is covered in appendix D.

### 3.3 ALIGNMENT WITH HUMAN PREFERENCES

**Datasets**   We utilize two datasets from OpenAI (Stiennon et al., 2022) to measure the alignment of text similarity metrics with human preferences. Both datasets contain machine-generated summaries for texts from the TL;DR dataset, CNN articles, and Daily Mail articles. The axis dataset has human crowdworkers score each summary on its coherence, coverage, accuracy, and overall effectiveness. The comparisons dataset has crowdworkers choose the more effective of two selected summaries.

**Evaluation**   For the axis dataset, we compute the similarity score between the machine-generated summary and the reference text. We then determine the correlation of these scores with the human scores in each scoring category.

For the comparisons dataset, we determine the similarity score between the two provided summaries and the reference text, setting the metric's choice to the summary given the higher similarity score. We judge each metric on its accuracy against the true choices made by the human labelers.

While the axis dataset has humans provide a discrete score between 1 and 10 for each category, we choose to frame this as a regression task and use a simple linear regression to develop an ensembled text similarity model, with our target values being the overall score for each document-summary pair. On the other hand, the target in the comparisons dataset is a binary variable, so we use a random forest classifier to develop our ensembled model.

Table 1: Metric performance on axis task

| Metric | Overall | Accuracy | Coverage | Coherence |
|---|---|---|---|---|
| Cosine | 0.64 | 0.61 | 0.65 | **0.63** |
| Levenshtein | 0.43 | 0.50 | 0.43 | 0.54 |
| Rouge | 0.53 | 0.53 | 0.53 | 0.55 |
| Jaccard | 0.55 | 0.55 | 0.55 | 0.55 |
| BM25+ | 0.64 | 0.58 | 0.65 | 0.54 |
| Ensembled model | **0.76** | **0.64** | **0.77** | 0.63 |

Table 2: Metric performance on comparisons task

| Metric | Accuracy | Precision | Recall | F1 score |
|---|---|---|---|---|
| Cosine | 0.63 | 0.62 | 0.63 | 0.62 |
| Levenshtein | 0.63 | 0.61 | 0.62 | 0.62 |
| Rouge | 0.63 | 0.60 | 0.65 | 0.63 |
| Jaccard | 0.63 | 0.61 | 0.65 | 0.63 |
| BM25+ | 0.63 | 0.60 | 0.66 | 0.63 |
| Ensembled model | **0.68** | **0.68** | **0.67** | **0.66** |

**Results**   The results for the axis task and comparisons task are presented in table 1 and table 2, respectively. Of the pre-existing metrics we explore, cosine similarity and BM25+ compete for first and second in the axis task with overall correlations of 0.64, with the remaining metrics lagging significantly. Levenshtein distance does especially poorly. In the comparisons task, our results have much less variance, with each metric scoring equally well with an accuracy of 0.63..

For both tasks, our ensembled models achieve a significant boost in performance. We improve correlation with human scoring to 0.76, an 18.8% improvement over the next best pre-existing metrics.

The only category where our ensembled model doesn't beat a pre-existing metric's score is coherence, where it ties with cosine similarity. In the comparisons task, we also see strong results from our ensembled model, achieving an overall accuracy of 0.68.

## 3.4 ROBUSTNESS TO SEMANTIC AND SUPERFICIAL TRANSFORMATIONS

**Datasets** Our robustness task necessitates datasets consisting of long-form content and human-generated summaries. To this end, we utilize a dataset of PubMed articles (Cohan et al., 2018), removing the abstract from the publication and using it as a summary. We also use datasets of CNN articles and government bills/proposals (Eidelman, 2019), both with human-generated summaries.

**Evaluation** In this task, we want to measure the ability of a text similarity metric to differentiate between superficial and semantically altering changes to a text. We create a set of transformations for each category.

**Superficial Transformations:**

- Random Capitalization: Randomly capitalizing 25% of letters in the text.
- Deletion: Removing every $10^{th}$ letter from the text, excluding spaces.
- Numerization: Substituting specific letters with numerals (e.g., replacing 'o' with '0').

Despite these changes, the semantic content of the text remains intact and a human would be able to recognize the similarity.

**Semantically Altering Transformations:**

- Negation: Changing affirmative statements to negative (e.g., changing "is" to "is not").
- Shuffle Sentences: Randomly rearranging the order of sentences.
- Shuffle Words: Scrambling the order of words within the text.

These changes disrupt the semantic structure of the original document. For example, shuffling the words in a text often renders it nonsensical. Humans can recognize that these transformations completely alter the original meaning of a text.

When evaluating each metric, we build on three assumptions.

- Superficial > Semantic: The similarity between text and a superficial transformation should be higher than with a semantically altering transformation.
- Summary > Semantic: The similarity between a text and its summary should be higher than with a semantically altering transformation.
- Superficial > Summary: The similarity between a text and a superficial transformation should be higher than with its summary.

To score a metric on these conditions, we determine whether the similarity scores given to a document and its transformations fulfill each condition above. When we measure against superficial or semantically altering transformations, we only assign a positive label to a datapoint if the condition holds for all transformations in each category.

When developing an ensembled model, we use a simple linear regression and train it on pairs of transformations. We assign a label of 0 when comparing a document and a semantic alteration, a label of 1 when comparing a document to a superficially transformed version, and a label of 0.5 when comparing a document and its summary (Ren & Li, 2023).

**Results** Our findings are shown in table 3. Surprisingly, ensembling achieves a near-perfect score of 0.98. This is likely due to our model being trained with an induced margin between the transformations and the summary, while the pre-existing metrics we use weren't created with this set of adversarial conditions in mind.

Table 3: Metric performance on robustness task

| Metric | Summary > Semantic | Superficial > Semantic | Superficial > Summary | Overall |
|---|---|---|---|---|
| Cosine | 0.06 | 0.00 | 0.55 | 0.21 |
| Levenshtein | 0.00 | 0.00 | 1.00 | 0.33 |
| Rouge | 0.00 | 0.00 | 0.71 | 0.24 |
| Jaccard | 0.00 | 0.00 | 0.60 | 0.20 |
| BM25+ | 0.63 | 0.18 | 0.27 | 0.36 |
| Ensembled | **0.97** | **0.98** | **1.00** | **0.99** |

## 3.5 SENSITIVITY OF MEASUREMENTS TO UNRELATED TEXT

**Datasets** When measuring a metric's performance on our sensitivity task, we minimize dataset bias by validating our results across a diverse array of documents. To accomplish this, we utilize datasets of PubMed publications (Cohan et al., 2018), essays written by thought leader Paul Graham (Goel, 2024), BBC News articles, Amazon reviews (Zhang et al., 2016), Reddit posts (Geigle et al., 2021), and the Argumentative Analysis (ArguAna) dataset (Wachsmuth et al., 2018). Our choice of datasets and evaluation methodology for this task are borrowed from (et. al, 2024).

**Evaluation** We want to determine the sensitivity of a text measurement to sentence-level changes in a multi-sentence document. We take two approaches. First, we add an irrelevant or adversarial piece of text ("needle") to a document (Guerreiro et al., 2023). with a varying needle's length (5%, 10%, 25%, 50%, and 100% of the original text's token count) and position (beginning, middle, end). Second, we instead remove chunks of text with varying sizes (5%, 10%, 25%, and 50%) and location. We compare the ablated text to the original using each metric, asserting that an ideal text similarity metric should decrease linearly as the perturbation size increases. For example, adding a needle that is 100% the length of the original text should decrease similarity by 50%, and removing 25% of the text should reduce similarity by 25%. This change should be agnostic to the position of the perturbation, so we average the error of each data point across all experimental conditions.

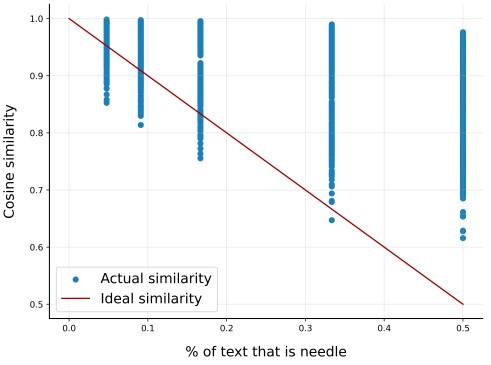
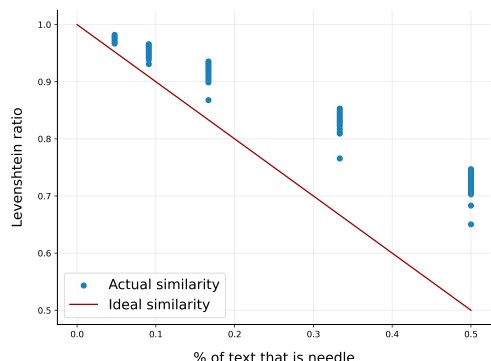

Figure 1: Ideal vs. actual cosine similarity    Figure 2: Ideal vs. actual Levenshtein ratio

To measure error, we compute the Mean Absolute Error (MAE) between each datapoint in each experimental condition and the behavior of our ideal metric, shown with an example set of measurements using BGE-M3 on the Paul Graham essay dataset in 1 and 2. Using MAE ensures that our error is bounded between 0 and 1, and so each metric's final score of $1 -$ error is also bounded.

We develop an ensembled model by training a simple linear regression to predict the ideal similarity score for each pairing between a document and its perturbed version using the corresponding scores from our pre-existing metrics.

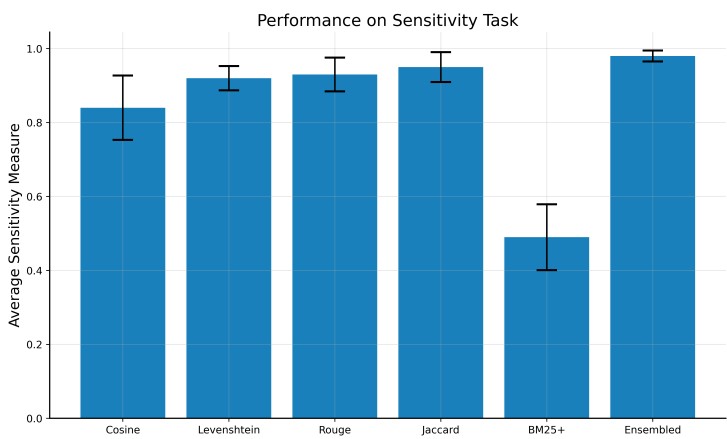

Figure 3: Metric performance on sensitivity tasks

**Results**   Our results, shown in figure 3, reveal a surprising ordering where BM25+ does extremely poorly with a 0.45 average score, and every other metric performs reasonably well, with scores above 0.85. Of the remaining pre-existing metrics, cosine similarity has the lowest score, and Jaccard similarity performs the best.

Ensembling our metrics achieves a near-perfect score of 0.98, improving upon cosine similarity by 13.9%. While the characterization of ideal metric behavior can be subjective, we re-ran our experiments using both quadratic and sublinear transformations of our ideal similarity curve and found similar orderings between our tested metrics.

### 3.6    CLUSTERING PERFORMANCE

**Datasets**   We mimic the setup of METB's clustering subtask, reusing all 11 datasets in the benchmark. These datasets consist of paragraphs or sentences from Arxiv, StackExchange, Medrxiv, Biorxiv, and Reddit. For each sentence or paragraph, there is a dataset-specific label that we use as our clustering target. The datasets are separated based on whether they consist of sentences or paragraphs (s2s or p2p).

**Evaluation**   Given a known number of labels, we employ an Agglomerative Clustering (Müllner, 2011) model to group our datapoints. We train this model with a distance matrix reflecting the similarity between each pair of datapoints, with a separate matrix for each metric. We subtract each similarity score from 1 to have each entry of our matrix represent distance. Each metric's performance is evaluated using v-measure (Rosenberg & Hirschberg, 2007), which doesn't depend on the specific cluster label and hence is invariant to a permutation of labels. This scoring choice is borrowed from the MTEB.

To train an ensembled model, we fit a random forest classifier by concatenating the distances computed by our metrics for each pair of datapoints and training using its correct label.

**Results**   Results are shown in figure 4. Cosine similarity performs the best amongst our pre-existing metrics by a fairly large margin, likely owing to its ability to represent semantic concepts in comparison to more algorithmic measurement methods. Ensembling all the metrics, however, once again archives the top score in this task, with a 6.3% improvement over cosine similarity. BM25+ received the lowest score, indicating that its decreased ability to understand content groupings could be a weakness for use in pre-processing during information retrieval pipelines.

### 3.7    RETRIEVAL PERFORMANCE AND ROBUSTNESS

**Datasets**   We base our retrieval task on the original 18 datasets in the BEIR benchmark, consisting of Wikipedia articles, Quora answers, scientific papers, and other high-quality documents.

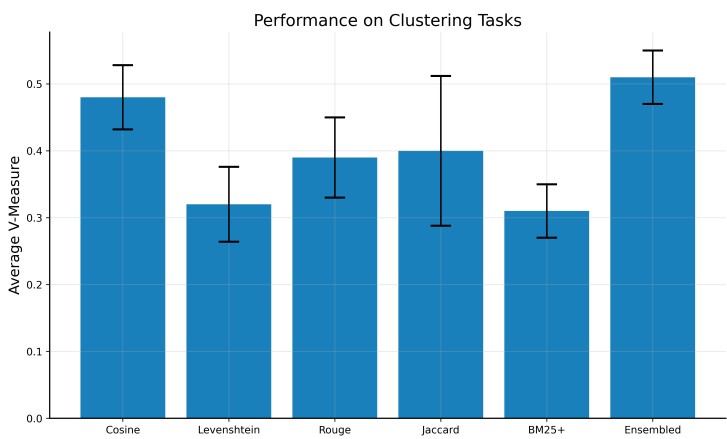

Figure 4: Metric performance on clustering tasks

**Evaluation** Our last task measures the ability of a metric to retain retrieval performance in unclean or adversarial environments, specifically in the context of information retrieval. To this end, we apply the transformations from 3.4 and 3.5 to each dataset, creating a perturbed copy of each datapoint for each transformation. We then compare the performance of a text similarity metric on BIER's retrieval tasks before and after the data augmentation and use the performance retained between the two scenarios as a measure of effectiveness. Our retrieval and scoring pipeline is described in appendix E.

To develop an ensembled measure, we train a random forest classification model on whether a document should or should not be selected by our retrieval pipeline.

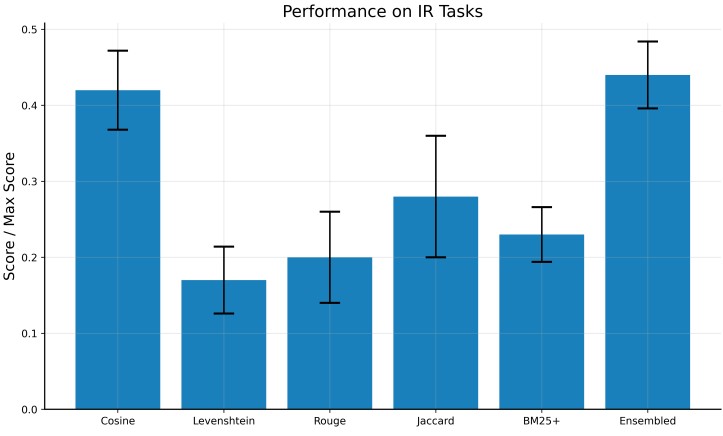

Figure 5: Metric performance on retrieval tasks

**Results** Cosine similarity scores the highest of any pre-existing text similarity metric, retaining 42% of its performance after dataset augmentation, This result is surprising given cosine similarity's performance in 3.4 and 3.5 but corroborates its position as a popular metric in retrieval tasks.

Ensembling sees minor gains in improvement, retaining 44% performance from the ensembled model's performance pre-augmentation, which itself achieved top results over pre-existing metrics.

## 4 INTRODUCING THE UNIFIED SEMANTIC SIMILARITY METRIC BENCHMARK (USMB)

We introduce the Unified semantic Similarity Metric Benchmark (USMB), our principal contribution designed to provide a comprehensive and standardized evaluation of text similarity metrics. The USMB aims to address the lack of a unified framework that assesses text similarity metrics across the multifaceted dimensions crucial for reliably measuring and quantifying semantic understanding. By integrating these diverse evaluation aspects, USMB offers a holistic leaderboard that not only ranks metrics based on their overall performance but also provides a useful demonstration of their strengths and weaknesses in specific tasks. This unique and novel approach is intended to guide practitioners in selecting the most appropriate metrics for their specific applications and to stimulate the development of more advanced similarity measures.

### 4.1 BENCHMARK DESIGN AND SCORING METHODOLOGY

The USMB is constructed by aggregating scores from the five unique tasks detailed in the previous sections. Each metric is evaluated on these tasks and assigned a normalized score between 0 and 1. The overall ranking is determined by averaging these scores.

Below, we detail the scoring methodology for each task included in USMB.

**Preference alignment**  We take the average of each metric's correlation with human scores on the axis dataset and its accuracy on human choice in the comparisons dataset. To avoid a negative score from a negative correlation, we adjust the correlation to be bounded between 0 and 1 with the following equation: $f(x) = 0.5 * (1 + x)$.

**Transformation robustness**  We use the average performance across all three semantically-motivated conditions laid out earlier.

**Information sensitivity**  The final score for a metric in this category is calculated by subtracting the MAE of each metric's outputs with our idealized similarity curve.

**Clustering performance**  We average the v-measure score of a metric across all datasets.

**Retrieval robustness**  We report the performance retention after data augmentation on retrieval tasks, framed as the retrieval score on the augmented task divided by the retrieval score on the original dataset.

### 4.2 BENCHMARK RESULTS AND ANALYSIS

Table 4 presents the USMB scores for each evaluated metric, including our proposed task-specific ensembled models.

Table 4: USMB benchmark scores

| Metric | Alignment | Robustness | Sensitivity | Clustering | Retrieval | Overall |
|---|---|---|---|---|---|---|
| Cosine similarity | 0.64 | 0.21 | 0.86 | 0.49 | 0.42 | 0.52 |
| Levenshtein ratio | 0.52 | 0.33 | 0.90 | 0.32 | 0.16 | 0.45 |
| Rouge score | 0.57 | 0.24 | 0.92 | 0.39 | 0.20 | 0.46 |
| Jaccard similarity | 0.58 | 0.20 | 0.96 | 0.41 | 0.28 | 0.49 |
| BM25+ score | 0.62 | 0.36 | 0.48 | 0.31 | 0.24 | 0.4 |
| Task-based ensembling | **0.72** | **0.98** | **0.98** | **0.51** | **0.44** | **0.73** |

Among the pre-existing metrics, cosine similarity achieves the highest overall score of 0.52. Its high sensitivity score of 0.86 corroborates the effectiveness of embedding models in capturing semantic meaning. However, its low robustness score of 0.21 suggests susceptibility to superficial textual alterations, which can be problematic in real-world applications involving noisy text data.

Jaccard similarity and Rouge score also perform well, coming in second and third place. There's no strong difference in relative scores between tasks when compared to cosine similarity however, with the sole exception being in the robustness task, indicating that token similarity can be a more powerful proxy than embeddings when it comes to capturing changes in texts that are more obvious to humans than LLMs. In line with our previous findings, our task-specific ensembled models perform the best across all tasks we measured with an overall score of 0.73, 40.3% higher than pure cosine similarity and 82.5% better than BM25+.

We see the biggest gain from ensembling in the robustness and sensitivity subtasks, where measuring semantic similarity is both the most subjective and the most difficult to discern for automated text similarity metrics. When we decompose the contribution of each metric to the ensemble model's final prediction, we find that no one metric dominates in its contributions. The contribution of each metric to the final ensemble model for each task is covered in appendix D.

## 5 DISCUSSION AND FUTURE WORK

The Unified semantic Similarity Metric Benchmark (USMB) represents a significant advancement in the ability to measure, evaluate, and select text similarity metrics in a way that is both robust and meaningful to their downstream usage. Our work introduces three key innovations: (1) a comprehensive, multi-task evaluation framework that spans a diverse set of commonly used metrics, (2) a novel ensemble approach that leverages lightweight machine learning models to improve task-based performance, and (3) a standardized methodology for comparing both traditional and neural similarity metrics. The USMB's multi-faceted and extensible design allows for tasks and metrics to be easily added, offering a new ground for text similarity metrics themselves to be put to the test and opening possibilities to quickly advance our ability to measure and capture semantic similarity.

Our decision to use out-of-the-box implementations with little testing of other models when ensembling underlines a key theme of our work: even simple ensemble methods can significantly improve the robustness and alignment of semantic similarity measurement. As far as we are aware, there are no other works that have systematically researched combining multiple text similarity methods for improving semantic understanding, and our work is additionally the first to create a suite of original tasks and datasets exclusively for evaluating text similarity measurements.

While our ensembled models exhibit remarkable performance on USMB tasks, we acknowledge certain limitations. The generalizability to unseen data and domains, computational complexity of the ensemble approach, and dependence on specific training datasets are areas that warrant further investigation. These challenges present exciting opportunities for future research and development in the field of text similarity metrics.

Looking ahead, there are several distinct directions of future work. Expanding the USMB to include domain-specific and multilingual datasets as part of its evaulation process can enhance its robustness and applicability. Putting more focus on other ensembling methods such as gradient-boosted trees or small neural networks can improve performance on each task even further. Lastly, it is a worthwhile endeavor to determine if ensembling text similarity metrics can generalize to strong performance across all tasks with the same model, instead of having a new ensembled model for each task.

## 6 CONCLUSION

We introduce the Unified semantic Similarity Measure Benchmark (USMB), a novel benchmark meant to measure the ability of text similarity measurements to reflect a human-level understanding of semantic similarity. We focus on 5 unique tasks: human preference alignment, transformation robustness, information sensitivity, clustering performance, and retrieval robustness. We discover that each text similarity metric we use exhibits varying strengths and weaknesses, and no metric dominates across all tasks. Thus, we develop and measure the performance of task-specific ensembled models using our pre-existing measurements and achieve top scores across all tasks, often times by a considerable margin. Our findings demonstrate that special care must be given to the selection of a text similarity measurement based on the task it is intended for, and that classical machine learning techniques can be used to augment the performance of existing text similarity measurements.

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

# A EQUATIONS FOR TEXT SIMILARITY MEASUREMENTS

## A.1 LEVENSHTEIN RATIO

$$\text{Levenshtein ratio}(str_1, str_2) = \frac{len(str_1) + len(str_2) - \text{Levenshtein Distance}(str_1, str_2)}{len(str_1) + len(str_2)}$$

## A.2 JACCARD SIMILARITY

$$\text{Jaccard similarity}(str_1, str_2) = \frac{|set(str_1) \cap set(str_2)|}{|set(str_1) \cup set(str_2)|}$$

## A.3 TF-IDF

$$\text{TF}(t, d, D) = \frac{\text{Number of times term } t \text{ appears in document } d}{\text{Total number of terms in document } d}$$

$$\text{IDF}(t, D) = \log\left(\frac{\text{Total number of documents } D}{\text{Number of documents with term } t \text{ in it}}\right)$$

$$\text{TF-IDF}(t, d) = \text{TF}(t, d) * \text{IDF}(t, D)$$

## A.4 BM25+

$$\text{BM25}(d, q, D) = \sum_{t \in q} \text{IDF}(t, D) \times \frac{\text{TF}(t, d) \times (k_1 + 1)}{\text{TF}(t, d) + k_1 \times \left(1 - b + b \times \frac{|d|}{\text{avgdl}}\right)} + \delta$$

## A.5 ROUGE SCORE

$$\text{Rouge}_n(str_1, str_2) = \frac{\sum_{\text{gram}_n \in str_1} \text{Count}_{str_2}(\text{gram}_n)}{\sum_{\text{gram}_n \in str_1} \text{Count}(\text{gram}_n)}$$

## A.6 COSINE SIMILARITY

$$\text{Cosine Similarity}(x, y) = \frac{dot(x, y)}{||x||_2 ||y||_2}$$

## B    MODEL DETAILS

**Text-Embedding-3-Small (OpenAI, 2024)**    has a content length of 8192 tokens and an unknown number of parameters. The model is accessed via the OpenAI API.

**Embed-English-v3.0 (Reimers, 2023)**    has a content length of 512 tokens and an unknown number of parameters. The model is accessed via the Cohere API.

**BGE-m3 (Chen et al., 2024)**    has a content length of 8912 tokens, is comprised of 568M parameters, and was trained using the APE positional encoding method.

**E5-Large-V2 (Wang et al., 2022)**    has a content length of 512 tokens, is comprised of 335M parameters, and was trained using the APE positional encoding method.

**Nomic-Embed-Text-v1.5 (Nussbaum et al., 2024)**    has a content length of 8192 tokens, is comprised of 137M parameters, and was trained using the RoPE positional encoding method.

**E5-RoPE-base (Zhu et al., 2024)**    has a content length of 512 tokens, is comprised of 108M parameters, and was trained using the RoPE positional encoding method.

**Jina-Embeddings-v2-Base (Günther et al., 2024)**    has a content length of 8192 tokens, is comprised of 137M parameters, and was trained using the ALiBi positional encoding method.

**Mosaic-Bert-Base (Press et al., 2022)**    has a content length of 1024 tokens, is comprised of 110M parameters, and was trained using the ALiBi positional encoding method.

## C    ENSEMBLE MODEL AND TEXT SIMILARITY METRIC COMPUTATIONAL COMPLEXITIES

The computational complexity of a trained ensemble method is 9 FLOPS for a linear regression model in inference mode. The model takes in 5 features, performs an element-wise multiplication with each of these features (5 FLOPS), then adds them together (4 FLOPS).

The random forest classification model used in our paper uses 9600 FLOPS for inference. Given 5 feature inputs and an estimated depth of 10, the number of comparisons performed is at most $\lfloor\sqrt{10}\rfloor * 2^5 = 96$. Since we use 100 decision tree estimators in our random forest, the amount of FLOPS used for a single inference is $100 * 96 = 9600$.

Given these numbers, the computational complexity of a trained ensemble method is fairly negligible.

Additionally the worst-case time complexity of all text similarity metrics used, shown in table 5, is equal to or better than cosine similarity.

Table 5: Time complexity of tested similarity metrics

| Metric | Cosine | Jaccard | Levenshtein | BM25+ | Rouge |
|---|---|---|---|---|---|
| Time Complexity | $O(n^2)$ | $O(n)$ | $O(n^2)$ | $O(n)$ | $O(n)$ |

## D    DECOMPOSITION OF TEXT SIMILARITY METRIC CONTRIBUTION TO TASK-SPECIFIC ENSEMBLE MODELS

The relative importance of each metric in ensembling is calculated differently for each model. For linear models, we calculate the proportion of each metric's beta coefficient against the sum of absolute beta coefficients. Weights for each task that utilizes a linear model are shown in table 6.

Table 6: Similarity metric coefficients in ensembling tasks utilizing linear models

| Metric | Cosine | Jaccard | Levenshtein | BM25+ | Rouge |
|---|---|---|---|---|---|
| Human Preference - Axis Task | 0.34 | 0.11 | 0.27 | 0.14 | 0.14 |
| Robustness | 0.01 | 0.31 | 0.29 | 0.04 | 0.36 |
| Sensitivity | 0.03 | 0.31 | 0.26 | 0.02 | 0.37 |

For random forest models, we calculated the Mean Decrease in Impurity (MDI) from each input feature to measure its relative importance to the final model. Weights for the human preference (comparisons) task are shown in table 7.

Table 7: Similarity metric coefficients in ensembling tasks utilizing random forest models

| Metric | Cosine | Jaccard | Levenshtein | BM25+ | Rouge |
|---|---|---|---|---|---|
| Human Preference - Comparisons Task | 0.34 | 0.11 | 0.27 | 0.14 | 0.14 |

# E   SETUP OF INFORMATION RETRIEVAL PIPELINE

Our information retrieval pipeline in the retrieval robustness task consists of the following steps:

1. Embed all documents into vectors and add them to a vector database.
2. Embed the query (search term) into a vector.
3. Return the top-$n$ documents with the highest similarity scores with the query.

After retrieving our top-$n$ documents, we determined how many matched our provided "ideal" documents to determine a metric's score as a percentage of the highest possible score it could achieve given perfect retrieval.

