# OpenReview forum: "Beyond Cosine Similarity: Introducing the Unified semantic Similarity Metric Benchmark (USMB) for Text Similarity Measurement"
_ICLR.cc/2025/Conference — Submitted to ICLR 2025_

### Official Review · Reviewer_zikP · 2024-11-01

**Soundness:** 1
**Presentation:** 2
**Contribution:** 2
**Rating:** 3
**Confidence:** 4

**Summary:**

This paper introduces the Unified Semantic Similarity Metric Benchmark (USMB), a benchmark for evaluating text similarity metrics across five tasks: human preference alignment, transformation robustness, information sensitivity, clustering performance, and retrieval robustness. The authors evaluate several pre-existing text similarity metrics like cosine similarity, Levenshtein distance, Rouge score, BM25, and Jaccard similarity on these tasks. They find that while cosine similarity achieves the highest overall score among the pre-existing metrics, no single metric dominates across all tasks. To improve performance, the authors develop task-specific ensemble models that combine these metrics. These ensemble models outperform all the pre-existing metrics.

**Strengths:**

The key contributions of this work are: (1) introducing an evaluation benchmark for text similarity metrics, (2) showing that ensembling multiple metrics can significantly boost performance on this benchmark. The paper does provide a good reminder that combining multiple types of similarity is likely to work better than cosine alone.

**Weaknesses:**

Novelty: The novelty is very limited, and the paper makes novelty claims that are not accurate: it is known that combining string-based similarity metrics with embedding-based ones is likely to give better results, and that is even more likely when there is training involved; this was extensively studied for example in the retrieval literature, where it is common to combine metrics based on sparse representations (eg bm25) with embedding based similarity.

Methodology:
* The paper looks at average scores across different models and draws conclusions regarding similarity metrics. However the performance variation across different models is very relevant: for example, is it the case that some models benefit more than other from metric aggregation? Perhaps strong embedding models benefit much less: this is an important question given that both embedding algorithms and similarity metrics try to solve the same problem. To pick the example of robustness: robustness can be addressed at the embedding level as well as at the similarity metric level. For this reason, the interaction with the embedding model chosen can’t be ignored.
* The benchmark is set up such that you train a metric on the train split of the benchmark and test it on the test portion. I would not trust this setup to make decisions about my metric because it may not generalize to new data or tasks. On the other hand if the downstream task I am targeting is clear to me and I do not need generalization, then I could use a task or a domain-specific setup to train a similarity metric.
* The three assumption in lines 250 to 256 seem pretty adhoc, especially the third one. An alternative is to chose a set of downstream tasks and simply test task performance under different perturbations.

Limitations of the benchmark:
The benchmark is limited in several dimensions, for example the task of alignment with human preferences is limited to comparisons between machine generated text and human text. Also it does not sound like the humans have access to the reference text, so they are basically solving a different task. Given the name of the task, I expected this to compare automatic similarity scores with human-assigned similarity scores. For the robustness tasks, the assumptions made in evaluating this metric are again limiting, and the data is restricted to summaries alone.

**Questions:**

here are several claims made throughout the paper that are simply not accurate. I would recommend the authors to re-consider such claims or back them up in revisions:
* lines 41-45: there is a sore lack of exploration into measuring semantic similarity. similarly in abstract: “relatively little research in how to best utilize embeddings for downstream tasks” or lines 511-516: “there are no other works that have systematically researched combining multiple text similarity metrics“: please see the extensive work done on this topic in machine translation evaluation or retrieval for example.
* presentation in section 2 is not clear: I recommend formalizing the problem, so that the metrics can be described accurately: for example TF-IDF introduces documents, but its not clear how that relates to comparing 2 arbitrary pieces of text. Also, it would help to clearly distinguish between string-based from embedding based metrics.

---

### Official Review · Reviewer_tCxV · 2024-11-03

**Soundness:** 3
**Presentation:** 3
**Contribution:** 3
**Rating:** 6
**Confidence:** 4

**Summary:**

The authors propose a novel benchmark for semantic text similarity. The benchmark consists of 5 semantic tasks with 30 datasets for the evaluation of similarity approaches. The main contributions are: i) semantic similarity benchmark, and ii) comprehensive evaluation of similarity approaches. The benchmark shows that ensemble methods outperform standard similarity metrics (e.g. cosine similarity).

**Strengths:**

- Benchmark for semantic text similarity with data and different baselines.
- Clear description of background knowledge and related work needed to understand the proposed benchmark and approaches.
- The authors perform a  comprehensive comparison of metrics and neural models with the proposed benchmark.

**Weaknesses:**

- Dependence on monolingual (e.g. English) datasets.

**Questions:**

Please address the following questions during the rebuttal:

- Please define the language or languages used by the benchmark.
- Elbarotate on possibilities to extend the benchmark into a multilingual setting.
- Please speculate if an evaluation under OOD data could be managed by the robustness section of the benchmark. For example, domain shift in the data.

Extra:
- In line 289 there is an error with the anonymous reference.

**Details Of Ethics Concerns:**

I have no concerns.

---

### Official Review · Reviewer_npqy · 2024-11-04

**Soundness:** 2
**Presentation:** 1
**Contribution:** 2
**Rating:** 3
**Confidence:** 4

**Summary:**

The paper explores ensembling different similarity measures and evaluates their performance across datasets designed to assess five aspects of semantic similarity: alignment with human judgment, robustness to text transformations, sensitivity to unrelated information, clustering effectiveness, and retrieval robustness. Focusing on text similarity, the authors employ multiple models to generate text embeddings throughout their experiments. By fitting task-specific ensembles for each category, the authors achieve higher scores compared to the individual similarity metrics. To standardize this approach, the authors propose a benchmark for semantic similarity metrics, outlining certain similarity aspects and relevant datasets for each category.

**Strengths:**

Combining different similarity metrics is promising, mainly as it seems to offer better alignment with human perceptions of similarity. While this isn’t explicitly addressed in the paper, it’s reasonable to hypothesize that each metric captures unique aspects of similarity, allowing for meaningful combinations.

The experiments are broad, covering multiple datasets and embedding models. They offer a valuable overview of the strengths and limitations of individual metrics, highlighting issues such as cosine similarity’s poor robustness.

**Weaknesses:**

The related work is superficial, mainly listing standard similarity metrics without providing much context. Moreover, the discussion omits relevant metrics geared toward similarity in neural network representations, such as CKA (Kornblith et al., 2019) and generalized shape metrics (Williams et al., 2021). These metrics should be included and compared in the experimental setup, as they represent significant approaches to semantic similarity, particularly relevant in the context of LLMs. Overall, the experiments lack stronger baselines.

The evaluation setup lacks proper grounding, as it involves comparing five standard similarity metrics against an ensemble specifically fitted to each dataset category. A key issue here is that the ensemble is directly trained on each dataset, meaning it has been optimized to perform well within the specific context of the data it’s evaluated on. This setup naturally leads to higher scores for the ensemble, as it is tuned to the particular nuances of the dataset, while the individual similarity metrics — such as cosine similarity, which is simply calculated as the cosine of the angle between two text embeddings — remain static and unoptimized for the dataset in question. To assess the practical usefulness of this ensembled metric, it should ideally be evaluated on entirely different datasets than those used for training. This approach would better reflect a real-world scenario, where in practice, one does not typically have access to labeled texts for fine-tuning a model to every new dataset. Evaluating it only on the datasets it has been fitted to limits the validity of any claims about its general performance or superiority over other metrics. Additionally, some critical aspects are overlooked, such as a comparison of the runtime and computational demands of the ensemble relative to individual metrics.

While most of the chosen aspects of similarity are reasonable, the approach to sensitivity raises some concerns. The authors state, for example, that "adding a needle that is 100% the length of the original text should decrease similarity by 50%, and removing 25% of the text should reduce similarity by 25%." This implies a linear relationship between similarity and text length, which may not be a desirable property for a similarity metric. For instance, summaries are often much shorter than the original texts yet should still yield high similarity scores if they effectively capture the main content. In such cases, the similarity score should depend more on the content and relevance of the “needle” rather than simply on its length.

Moreover, this approach does not distinguish between scenarios where the added or removed text may be semantically unrelated, redundant, or adversarial, all of which could influence similarity differently. Without further clarification, this linear assumption risks oversimplifying the complexity of semantic similarity, as it neglects the roles of content and context in determining meaningful similarity scores.

The wording is sometimes imprecise and difficult to parse. For example, lines 105–107 could mislead readers into thinking that RAG is the only application of information retrieval in LLMs. Similarly, lines 178–180 are confusing due to unclear phrasing, making it difficult for readers to grasp the intended meaning.

The proposed benchmark feels like an afterthought, only mentioned toward the end. Additionally, the title seems misaligned with the paper's primary focus, which centers more on the empirical evaluation of standard similarity metrics compared to their ensembled versions. Overall, the paper lacks substance, as it offers no significant contribution. The practical utility of the ensembled metric is unclear in this evaluation setup. While the benchmark and the selected aspects of similarity are reasonable and appear relevant, they lack theoretical grounding and clear motivation.

**Questions:**

- How are categorical decisions determined when using similarity metrics that output a continuous score between 0 and 1?
- What exactly is the content of the "needle"? Are there multiple types, as suggested by the phrase, "...we add an irrelevant or adversarial piece of text ('needle')..."?

---

### Official Review · Reviewer_Knn5 · 2024-11-04

**Soundness:** 3
**Presentation:** 3
**Contribution:** 2
**Rating:** 5
**Confidence:** 4

**Summary:**

- The authors focus on the widespread use of cosine similarity between text embeddings in various NLP-related tasks and attempt to validate the appropriateness of this approach across diverse datasets and task settings.
- Validation on existing datasets and tasks, as well as on new datasets with added perturbations, showed that ensembling simple metrics like cosine similarity and BM25 yields better performance than using cosine similarity alone.

**Strengths:**

- This paper tackles an appealing problem. Cosine similarity is widely used, yet the reasons for its effectiveness, especially from a theoretical perspective, remain unclear. New findings on this problem could provide valuable information for many NLP/ML practitioners. Generally, papers that carefully question de facto approaches tend to be of interest to both researchers and practitioners, and this paper follows this line.
- The experimental setup of adding perturbations to existing datasets is interesting. In particular, the figures, such as Fig. 1 and 2, are novel to me, and they are likely to offer valuable insights to many researchers in textual similarity.

**Weaknesses:**

The contributions are all marginal, making it difficult to say that the work meets the quality expected for acceptance at a top conference in representation learning.

- The empirical takeaway can be summed up as “ensembling improves scores,” which is somewhat trivial.
- The authors claim to introduce a benchmark, but in reality, they simply use several existing datasets in their experiments. The proposal to add perturbations is appealing, and the semi-automated generation of parts of the summarization dataset is also a contribution. However, describing this as a benchmark proposal seems somewhat of an overclaim.
- There is no theoretical contribution. If an ICLR reader were to read the first paragraph of this paper, they would likely expect some theoretical implications regarding the strengths or weaknesses of cosine similarity. Building on existing work, such as Zhelezniak et al. NAACL 2019 (arXiv: 1905.07790), to offer new insights would enhance the appeal of this paper.

**Questions:**

- It seems that the validity of using cosine similarity for test-time evaluation might depend on which metric—cosine, inner product, or L2—is used during the training of text embedding models, particularly in contrastive learning. In §3.1, many models are mentioned, but are there any implications in the experimental results regarding the training-test discrepancy?
- The proposal to add perturbations is interesting, but the assumption that semantic similarity decreases linearly with the degree of perturbation seems somewhat simplistic. Is this linearity actually observed across a large number of specific texts? If not, are there any solutions to address this?

---

### Comment · Area_Chair_6JXR · 2024-11-21
**Reminder: Please respond and update the score if necessary**

Dear Reviewers,

Kindly ensure that you respond proactively to the authors' replies (once they are available) so we can foster a productive discussion. If necessary, please update your score accordingly. We greatly appreciate the time and effort you’ve dedicated to the review process, and your contributions are key to making this process run smoothly.

Thank you,

AC

---

### Meta-Review · Area_Chair_6JXR · 2024-12-19

**Metareview:**

The paper investigates the effectiveness of cosine similarity in NLP tasks and examines the benefits of combining different similarity measures. Through their research, the authors demonstrate that ensembling simple metrics like cosine similarity and BM25 often outperforms using cosine similarity alone. They introduce the Unified Semantic Similarity Metric Benchmark (USMB) to evaluate text similarity metrics across five areas: human preference alignment, transformation robustness, information sensitivity, clustering performance, and retrieval robustness. The study finds that while cosine similarity performs well overall, no single metric is best for all tasks. By developing task-specific ensemble models, they achieve superior performance compared to individual metrics.

The reviewers commend the paper for its significant contributions, including the creation of a new evaluation benchmark for text similarity metrics and the finding that combining multiple metrics greatly improves performance. The paper serves as a strong reminder that integrating various similarity measures often outperforms using cosine similarity alone. However, the paper lacks theoretical insights and shallow analysis of cosine similarity. Additionally, the novelty is limited, as it is already well-known that combining string-based and embedding-based similarity metrics enhances performance, a concept extensively studied in the retrieval literature.

I recommend rejecting the paper because it lacks theoretical insights and offers limited novelty, as noted by the reviewers. Although the results are intriguing, a more robust analysis is necessary to fully understand the findings presented, including the effectiveness of the proposed similarity measures.

**Additional Comments On Reviewer Discussion:**

No progress has been made, as neither the reviewers nor the authors have participated in any follow-up discussions. I've urged both parties to initiate discussions during the author response period.

Reviewer zikP conducted a comprehensive review, pointing out critical issues with the experiment setup, which is a major weakness of this work. Additionally, Reviewer Knn5 emphasized the lack of a theoretical foundation, which I believe is essential for an ICLR conference paper to sufficiently support its experimental results.

---

### Decision · Program_Chairs · 2025-01-22

Reject